# Enhancing Vision-Language Model Reliability with Uncertainty-Guided Dropout Decoding

**Yixiong Fang**[*]
Carnegie Mellon University
yixiongf@cs.cmu.edu

**Ziran Yang**[*]
Princeton University
zirany@princeton.edu

**Zhaorun Chen**
University of Chicago
zhaorun@uchicago.edu

**Zhuokai Zhao**[†]
University of Chicago
zhuokai@uchicago.edu

**Jiawei Zhou**[†]
Stony Brook University
jiawei.zhou.1@stonybrook.edu

## Abstract

Large vision-language models (LVLMs) excel at multimodal tasks but are prone to misinterpreting visual inputs, often resulting in hallucinations and unreliable outputs. We present DROPOUT DECODING, a novel inference-time approach that quantifies the uncertainty of visual tokens and selectively masks uncertain tokens to improve decoding. Our method measures the uncertainty of each visual token by projecting it onto the text space and decomposing it into *aleatoric* and *epistemic* components. Specifically, we focus on *epistemic* uncertainty, which captures perception-related errors more effectively. Inspired by dropout regularization, we introduce uncertainty-guided *token dropout*, which applies the dropout principle to input visual tokens instead of model parameters, and during inference rather than training. By aggregating predictions from an ensemble of masked decoding contexts, we can robustly mitigate errors arising from visual token misinterpretations. Evaluations on benchmarks including CHAIR, THRONE, and MMBench demonstrate that DROPOUT DECODING significantly reduces object hallucinations (OH) and enhances both reliability and quality of LVLM outputs across diverse visual contexts. Code is released at https://github.com/kigb/DropoutDecoding.

## 1 Introduction

Recent advancements in large vision-language models (LVLMs) have demonstrated impressive capabilities [1, 2, 3] in tasks such as image captioning, visual question answering (VQA), and multimodal reasoning [4, 5, 6, 7]. However, LVLMs still face challenges in accurately perceiving and interpreting visual inputs, leading to inaccurate outputs and hallucinations [8]. These issues often stem from LVLMs misrepresenting key image elements or overlooking critical details [9, 10]. In practice, LVLMs typically process visual inputs token by token [11], which we refer to as *visual tokens*.[3] This can fall short in effectively focusing on the most informative parts of the visual context. While attention mechanisms are designed to prioritize relevant information, they are not always perfect [12, 13, 14], especially when the inputs are complex or ambiguous for the model, or in other words, of high *uncertainty*. Existing methods to address these challenges in the training stage often involve fine-tuning on specific tasks [15, 16, 17, 18, 19], or using additional supervision signals especially at lower level to guide the model [20, 21]. However, these approaches are resource-intensive

---

[*]Equal contribution. Work done during their research internship at Stony Brook University.

[†]Joint last author.

[3]We specifically refer to the tokens that are already in the input prompt to the text decoder. Concrete definition is in §3.1.

39th Conference on Neural Information Processing Systems (NeurIPS 2025).

and not easily extensible to new tasks. Alternative inference-time strategies rely on attention or logits-based mechanisms but typically use heuristic designs and increase inference cost [22, 23, 24, 25]. Therefore, enhancing the trustworthiness of LVLMs [26] and reducing hallucinations [24] require more principled methods that can more effectively emphasize the most informative parts of the visual input.

To address this challenge, we propose a novel approach that quantifies uncertainty in visual token contexts and removes uncertain tokens, both directly at inference time to improve the reliability of LVLM outputs. Inspired by traditional dropout [27] techniques—typically applied to model parameters but difficult to implement directly in pretrained LVLMs [28, 29]—we introduce *token dropout*, which applies the dropout principle to input context tokens instead of model parameters. Furthermore, it is applied to regularize the inference process instead of training, by introducing randomness in decoding contexts to reduce overfitting to noisy visual tokens.

Our method measures the uncertainty of each visual token by projecting it into the text token space *through the text decoder directly*, and decomposing this uncertainty into two components: *aleatoric* (data-related) and *epistemic* (model-related) [30, 31, 32]. By focusing on epistemic uncertainty, which reflects the model's lack of knowledge, we identify visual tokens with high uncertainty and selectively target them for suppression. At inference time, we adjust the visual inputs by selectively suppressing tokens with high epistemic uncertainty. Specifically, we create an *ensemble of predictions* by generating multiple subsets of visual inputs, each with different combinations of high-uncertainty tokens dropped out. These subsets are processed independently, and their corresponding outputs are aggregated using majority voting to produce the final prediction.

Our method, termed DROPOUT DECODING, enhances the reliability and accuracy of LVLM outputs without modifying the underlying model parameters or requiring additional training. Experiments on LVLM decoding benchmarks including CHAIR [33], THRONE [34], and MMBench [35] demonstrate the effectiveness of our approach. In summary, we make the following contributions. First, we introduce a novel approach that quantifies and decomposes uncertainty on tokens in the visual inputs at inference time without additional supervision, by projecting visual input tokens onto text token interpretations. Second, we propose a decoding strategy that uses epistemic uncertainty measurements to guide the selective dropout of high-uncertainty visual tokens in the context, analogous to performing dropout on the model but applied to the input tokens and during inference. And finally, comprehensive experiments are conducted on various benchmarks, showing significant reductions in OH and improved fidelity in pre-trained LVLMs without additional fine-tuning.

## 2    Related Work

**Reliable Generation.** Hallucinations in LLMs—where models generate irrelevant or incorrect information [36, 37, 38]—arise from data [19], training, and inference issues [39], with attention mechanisms exacerbating them [40]. To address this, factual-nucleus sampling [41] balances diversity and accuracy. While [42] guide decoding with quantified uncertainty, our approach quantifies uncertainty at the visual input level, not requiring model ensembles.

**OH in LVLMs.** Object hallucination (OH) is common in LVLMs, where models generate incorrect object descriptions. CHAIR [33] and POPE [43] evaluate OH, while THRONE [34] offers a more holistic approach. We use CHAIR and THRONE to assess OH in our work.

**OH Reduction.** Methods addressing OH in LVLMs include internal signal guidance (e.g., OPERA [44]), contrastive decoding (e.g., VCD [45]), and selective information focusing (e.g., HALC [24]). AGLA [46] mitigates hallucinations by enhancing visual grounding through global and local attention, while Memory-Space Visual Retracing [47] refines multimodal alignment via iterative visual reference retrieval. In contrast, DROPOUT DECODING 1) selects visual tokens during generation, 2) uses uncertainty for token selection without external models, and 3) employs a token-level majority voting strategy.

## 3    Preliminaries

### 3.1    Vision-Language Model Decoding

Widely adopted LVLM architectures [48, 49, 17] typically include a vision encoder, a vision-text interface module, and a Transformer-based LLM decoder. As we mostly focus on the decoder side inference, we assume the LLM decoder parameterized by $\theta$.

The visual input, such as an image, is segmented into patches and processed by the vision encoder,[4] followed by the vision-text interface module, to produce a sequence of *visual tokens* $x^v = (x_1^v, x_2^v, \ldots, x_N^v)$. Each token $x_i^v$ is a contextualized embedding of an image patch, serving as the direct input to the text decoder. The text input such as a query or instruction is $x^t = (x_1^t, x_2^t, \ldots, x_M^t)$. The input to the text decoder is denoted as $x = [x^v, x^t]$, which is the concatenation of visual and text tokens. At this point, the visual and text tokens are aligned and serve as a sequential input to the LLM decoder. During autoregressive decoding, the decoder generates output text tokens $y = (y_1, y_2, \ldots)$ as continuation from prompt $x$, following the conditional probability distribution

$$h_j = f_\theta(x^v, x^t, y_{<j}), \quad p_\theta(y_j \mid x^v, x^t, y_{<j}) = \text{softmax}(W_\mathcal{V} h_j) \tag{1}$$

where $y_{<j} = (y_1, \ldots, y_{j-1})$ is the sequence of previously generated tokens, $f_\theta$ denotes the LLM forward pass to produce hidden states $h_j \in \mathbb{R}^d$ on top of the Transformer layers, $W_\mathcal{V} \in \mathbb{R}^{|\mathcal{V}| \times d}$ is the output projection matrix onto the text vocabulary $\mathcal{V}$, and $y_j \in \mathcal{V}$ the output token at $j$-th step.

### 3.2 Uncertainty Quantification

Our approach quantifies the information uncertainty of visual tokens used for decoding by adapting the concept of epistemic uncertainty for measurement, as detailed in §5, and drawing inspiration from classical uncertainty decomposition [31, 50, 32]. To provide the necessary background, we first introduce the concept of uncertainty decomposition.

Uncertainty decomposition separates the total uncertainty of a model's prediction into two components: *aleatoric* uncertainty, which is inherent to the data, and *epistemic* uncertainty, which relates to the model's lack of knowledge. The Bayesian framework offers a principled way to quantify uncertainty about some candidate model with weights $w$, through the posterior estimation over the hypothesis space for a given dataset $\mathcal{D}$. The Bayesian model average (BMA) predictive distribution is defined as[5]

$$p(y \mid x, \mathcal{D}) = \int_w p(y \mid x, w) p(w \mid \mathcal{D}) \, dw. \tag{2}$$

The total information uncertainty is measured by the entropy of BMA: $\mathbb{H}[p(y \mid x, \mathcal{D})]$, which equals the posterior expectation of the cross-entropy between the predictive distribution of the candidate model and the BMA distribution:

$$\underbrace{\mathbb{H}[p(y \mid x, \mathcal{D})]}_{\text{Total Uncertainty}} = \mathbb{E}_{p(w|\mathcal{D})} \left[ \text{CE}[p(y \mid x, w), p(y \mid x, \mathcal{D})] \right]$$

$$= \underbrace{\mathbb{E}_{p(w|\mathcal{D})} \left[ \mathbb{H}(p(y \mid x, w)) \right]}_{\text{Aleatoric Uncertainty}} + \underbrace{\mathbb{E}_{p(w|\mathcal{D})} \left[ D_{\text{KL}}(p(y \mid x, w) \parallel p(y \mid x, \mathcal{D})) \right]}_{\text{Epistemic Uncertainty}}$$

The epistemic uncertainty, expressed as the KL divergence between candidate models' predictive distributions and the BMA, has proven effective in various applications [51, 52, 28, 53]. Our approach, adopts a similar formulation for uncertainty quantification, calculating the KL divergence between candidate prediction distributions on individual visual tokens and an aggregated average distribution.

## 4 Textual Interpretation of Visual Tokens

As discussed in §1, identifying the visual tokens that carry significant information and quantifying their uncertainty is critical for improving the reliability of LVLMs. We propose a supervision-free approach that maps visual tokens to text token space for improving LVLM reliability by identifying significant visual tokens and quantifying their uncertainty. This mapping leverages the LVLM's inherent ability to align visual and textual contexts.

**Text-space projection of visual tokens.** While LVLMs are trained to generate text *only after* processing all visual tokens $x^v$ and text instruction tokens $x^t$, the hidden representations $h$ on top of the text decoder layers inherently capture textual semantics. This is due to their proximity to the text vocabulary projection, even at visual token positions where the model is *not explicitly trained to generate text*.

---

[4]We assume a general Transformer architecture for the vision encoder as well. Our approach could also apply to other types of vision encoders.

[5]$p(y \mid x, w, \mathcal{D}) = p(y \mid x, w)$ because of conditional independence.

Building on this intuition, we adopt a heuristic approach to interpret visual tokens by projecting them onto the text vocabulary at the top Transformer layers. In particular, for each visual token $x_i^v$ at position $i$,[6] we obtain its textual projected distribution over the vocabulary $\mathcal{V}$ from the last layer of the LLM decoder in the LVLM as:

$$h_i^v = f_\theta(x_{\leq i}^v)$$
$$q_i^{\text{proj}} = p_\theta(\cdot \mid x_{\leq i}^v) = \text{softmax}(W_\mathcal{V} h_i^v) \tag{3}$$

where $h_i^v$ is the LLM decoder top-layer hidden representation aligned at the $i$-th visual token positions, $x_{\leq i}^v$ denotes the visual tokens up until index $i$.[7] This approach is also generally referred to as logit lens [54] in mechanistic interpretability for LLMs.

Here, $q_i^{\text{proj}}$, which we refer to as *visual-textual distribution*, represents the projection of the visual input onto the text space. It encapsulates the model's interpretation of the $i$-th visual token. This projection offers a text-based summarization, akin to an unordered caption or a "bag-of-words" [15] representation of the visual content. As we will demonstrate in §6, this heuristic method serves as an effective proxy for uncertainty estimation.

**An illustrative example with projection uncertainty.** Figure 4 demonstrates our projection method by processing an image into patches and projecting five selected patches into the text space, retrieving their top-5 text tokens. Informative patches yield specific tokens like "*Berlin*," "*computer*," or "*map*," which are less frequent in the vocabulary and capture unique visual contexts. In contrast, patches producing common words (e.g., "*a*," "*the*," "*on*") convey less specific information. This suggests that projected text tokens effectively proxy the information content of visual tokens.

Leveraging this, we introduce uncertainty measures from the textual projection distributions $q_i^{\text{proj}}$ to quantify each visual token's uncertainty, as depicted in the figure. Following classical uncertainty quantification (§3.2), we decompose total uncertainty into *aleatoric* (data-related) derived directly from $q_i^{\text{proj}}$, and *epistemic* (model-related) by comparing $q_i^{\text{proj}}$ to an average distribution (§5.1). As illustrated, epistemic uncertainty aligns well with the information content of visual tokens: high epistemic uncertainty corresponds to informative patches (e.g., "*Berlin*"), and vice versa (e.g., "*the*"). In contrast, aleatoric and total

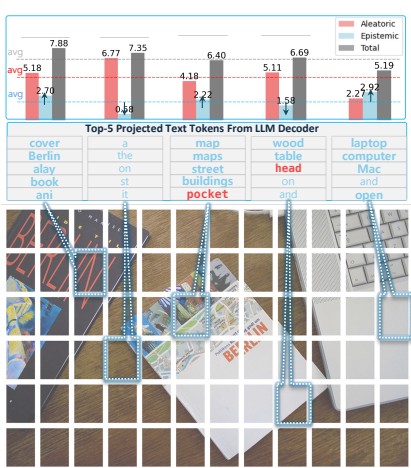

Figure 1: An illustrative example where visual tokens are projected into the text space, **bold** words indicate highly informative projections, and red words mark misalignments. Dotted lines show average uncertainties; high epistemic uncertainty correlates with informative patches.

uncertainty do not show this correlation. This finding motivates our focus on epistemic uncertainty as a reliable indicator of the significance of visual information.

## 5 Method

We propose DROPOUT DECODING, which leverages visual uncertainty to selectively drop out visual tokens and guide decoding. As shown in Fig. 2 and Algorithm 1, our approach comprises two stages: uncertainty quantification (§5.1) before decoding and uncertainty-guided token generation (§5.2) for decoding.

### 5.1 Uncertainty Quantification Before Decoding

**Average visual-textual distribution.** We begin by defining the averaged distribution $q^{\text{proj}}$, which represents the overall projection of the entire visual input (e.g. an image) into the text space. Using the projected distribution defined in Eq. (3), we define the average projection distribution over all

---

[6]Note that $i$ indexes are only used over visual tokens $x^v$, not text tokens $x^t$ or generations $y$.

[7]For the models we use, the visual tokens $x^v$ are all placed before the text tokens $x^t$ in the concatenated sequence $x$, so $x_{\leq i}^v$ are purely visual tokens. But our approach also applies to other cases.

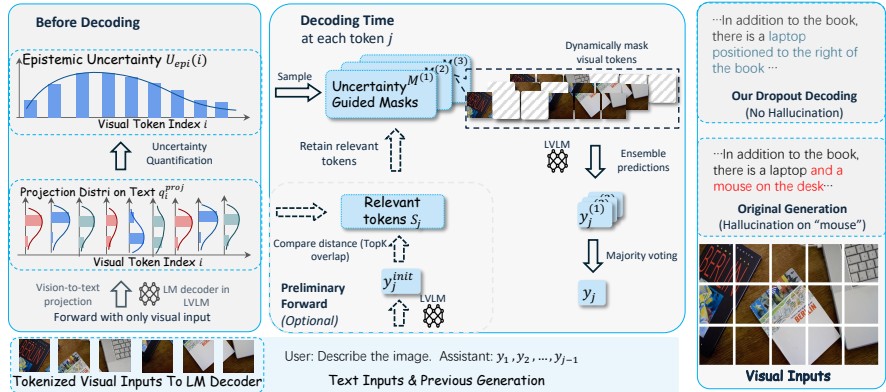

Figure 2: An overview of our DROPOUT DECODING. The method includes uncertainty measurement of visual tokens (under "*Before Decoding*") and uncertainty-guided visual context dropout decoding algorithm (under "*Decoding Time*"). The pseudocode is in Algorithm 1.

visual tokens as:

$$q^{\text{proj}} = \mathbb{E}_i[q_i^{\text{proj}}] = \frac{1}{N} \sum_i^N q_i^{\text{proj}} \tag{4}$$

where $q_i^{\text{proj}}$ represents the text-space projection of the $i$-th visual token, and $N$ is the total number of visual tokens. Note that the subscript $i$ indicates different distributions rather than elements within a single distribution. This provides us with a "baseline" representation of the visual input, against which we can quantify the surprisal of a specific visual token. This idea is grounded in classical uncertainty decomposition where a Bayesian average distribution is needed to quantify epistemic uncertainty [31, 32].

**Uncertainty measurement for visual tokens.** We aim to quantify the uncertainty associated with each visual token at inference time. To distinguish from those uncertainty terms in classical settings as introduced in §3.2, we refer to ours as *perception uncertainty*. We start by quantifying the *perception total uncertainty* of the visual input as the entropy of the average visual-textual distribution $\mathbb{H}\left[q^{\text{proj}}\right]$. Then, to attribute this total uncertainty to individual visual tokens, we decompose it (details in Appendix A) by:

$$U_{\text{total}} = \mathbb{H}\left[q^{\text{proj}}\right] = \mathbb{E}_i\left[\text{CE}\left(q_i^{\text{proj}}, q^{\text{proj}}\right)\right] \tag{5}$$

Further decomposing the cross-entropy (CE), the perception total uncertainty can be expressed as:

$$U_{\text{total}} = \mathbb{E}_i\left[\mathbb{H}\left[q_i^{\text{proj}}\right] + D_{\text{KL}}\left(q_i^{\text{proj}} \parallel q^{\text{proj}}\right)\right]$$
$$= \mathbb{E}_i\left[U_{\text{ale}}(i) + U_{\text{epi}}(i)\right]$$

Here we have the *perception **aleatoric** uncertainty* of the $i$-th visual token $U_{\text{ale}}(i) = \mathbb{H}\left[q_i^{\text{proj}}\right]$, capturing the inherent noise or ambiguity of the $i$-th token, and the *perception **epistemic** uncertainty*—

$$U_{\text{epi}}(i) = D_{\text{KL}}\left(q_i^{\text{proj}} \parallel q^{\text{proj}}\right) \tag{6}$$

quantifying the divergence between the visual token's textual projection and the overall projection. It indicates how much the model's belief about this token differs from its belief about the entire visual input. A higher $U_{\text{epi}}(i)$ suggests that the $i$-th visual token conveys information that is surprising or not well-represented in the overall visual content, which can be critical for identifying tokens that might introduce uncertainty in the decoding process.

## 5.2 Uncertainty-Guided Decoding

During the text decoding process, we leverage the computed uncertainty measures to guide the generation of each token. Our method involves two main steps for each generated *text* token: (1) identifying relevant visual tokens (optional), and (2) performing *token dropout* with uncertainty-guided masking. The first step is optional, designed to enhance decoding by retaining more relevant visual tokens.

**Identifying relevant visual tokens (optional).** We selectively retain only the most relevant visual tokens from the context, which are excluded for dropout. When generating each output text token, $y_j$, we first perform a preliminary forward pass to generate an initial prediction token $y_j^{\text{init}}$:

$$y_j^{\text{init}} \sim p_\theta(\cdot \mid x^v, x^t, y_{<j}) \tag{7}$$

Next, we determine the set of visual tokens that are relevant to this initial prediction. Specifically, a visual token $x_i^v$ is considered relevant if the initial prediction $y_j^{\text{init}}$ appears among the top-$k$ tokens of its visual-textual projection $q_i^{\text{proj}}$. Formally, the set of relevant visual tokens for the $j$-th generation is:

$$\mathcal{S}_j = \left\{ x_i^v \;\middle|\; y_j^{\text{init}} \in \text{TopK}(q_i^{\text{proj}}) \right\} \tag{8}$$

where $\text{TopK}(\cdot)$ denotes the function returning the top-$k$ entries of a given distribution.

To illustrate the intuition behind this step, consider an image depicting a cat. Suppose the model correctly predicts the token "*cat*" during the preliminary forward pass. In that case we retain the visual tokens associated with "*cat*" and drop out among the remaining visual content. Conversely, if the model incorrectly predicts "*dog*" or unrelated tokens irrelevant to an object, these predictions will not align with the top text projections of any $q_i^{\text{proj}}$ if the visual interpretation is accurate. In such cases, no visual tokens are retained due to a lack of clear relevance, and dropout is applied across the entire visual context as the best alternative.

It is worth noting that this step is optional. Omitting it can improve efficiency by reducing the computational overhead of the preliminary forward pass. As shown by the ablation studies in §7, while skipping this step may lower performance on certain benchmarks like THRONE [34], it still achieves comparable results on others such as CHAIR [33].

**Visual token dropout with uncertainty guidance.** Using the epistemic uncertainty measurements $U_{\text{epi}}(i)$ from Eq. (6), we introduce dropout masks over visual tokens. As illustrated in Fig. 4, the projected visual-textual distributions sometimes misalign with the image content, and regions of high information can lead to substantial errors, resulting in hallucinations. Based on this intuition, we selectively target visual tokens with high epistemic uncertainties for dropout.

Specifically, we formulate a controllable series of sample distributions for visual token dropout based on $U_{\text{epi}}(i)$, for each visual position $i$:

$$P_{\text{dropout}}^{(k)}(x_i^v) = \gamma^{(k)} \left( \frac{U_{\text{epi}}(i) - U_{\text{epi}}^{\text{min}}}{U_{\text{epi}}^{\text{max}} - U_{\text{epi}}^{\text{min}}} \right) + \delta^{(k)} \tag{9}$$

where $U_{\text{epi}}^{\text{min}}$, $U_{\text{epi}}^{\text{max}}$ are the minimum and maximum epistemic uncertainty values across all visual tokens, and $\gamma^{(k)}$ and $\delta^{(k)}$ are hyperparameters controlling the probability range of the dropout. By adjusting the values of $\gamma^{(k)}$ and $\delta^{(k)}$, we can modulate the intensity of visual token dropout. For further discussion on hyperparameters, see §7.2.

With the dropout distributions, we can sample dropout masks for each visual token independently. Denote the binary mask as $M^{(k)} \in \{0,1\}^N$, consisting of a binary indicator $M_i^{(k)}$ for each visual token $x_i^v$, where the conrresponding visual token is retained if $M_i^{(k)} = 1$, and dropped if $M_i^{(k)} = 0$. The dropout mask sampling follows $P(M_i^{(k)} = 0) = P_{\text{dropout}}^{(k)}(x_i^v)$, and the sampling is done for each visual token position independently. A higher value of $P_{\text{dropout}}(x_i^v)$ indicates that $x_i^v$ is more likely to be dropped out. If we performed the optional preliminary forward pass to identify relevant visual token set $\mathcal{S}_j$, these visual tokens are never dropped, i.e., $\forall x_i^v \in \mathcal{S}_j$, set $M_i^{(k)} = 1$ directly.

**Ensemble-based reliable generation.** Our inference-time context dropout introduces stochasticity, so we employ an ensemble decoding approach by independently sampling $K$ distinct dropout masks, $\{M^{(k)}\}_{k=1}^K$, to enhance generation quality. Since the masks are independent, the text generative distribution from $K$ masks can be efficiently computed in a parallel forward pass

$$y_j^{(k)} \overset{\text{Decoding}}{\sim} p_\theta(\cdot \mid x_{/M^{(k)}}^v, x^t, y_{<j}) \tag{10}$$

where $x_{/M^{(k)}}^v$ denotes the visual tokens after applying dropout mask $M^{(k)}$, and $\overset{\text{Decoding}}{\sim}$ denotes invariance to the decoding algorithm used (e.g., greedy search in our implementation, though others are applicable).

---
**Algorithm 1** Pseudocode of DROPOUT DECODING.
---
1: **Input:** visual tokens $x^v$, Text tokens $x^t$, Number of dropout masks $K$, Generation length $L$
2: **Output:** Generated sequence $y$
3:
4: **Before Decoding:**
5: Obtain visual text projecting distributions $q_i^{\text{proj}}$. {Eq (3)}
6: Compute average distribution $q^{\text{proj}}$. {Eq. (4)}
7: Compute epistemic uncertainty $U_{\text{epi}}(i)$. {Eq. (6)}
8: **for** $j = 1$ to $L$ **do**
9:     **Identifying relevant visual tokens (optional):**
10:     Generate preliminary token $y_j^{\text{init}}$. {Eq. (7)}
11:     Get relevant tokens $\mathcal{S}_j$ with $y_j^{\text{init}}$ and $q_i^{\text{proj}}$. {Eq. (8)}
12:     **Visual token dropout with uncertainty-guidance:**
13:     Get $K$ dropout prob $P^{(k)}$ with $U_{\text{epi}}(i)$. {Eq. (9)}
14:     Generate $K$ dropout masks $M^{(k)}$ based on $P^{(k)}$ while retain relevant tokens $\mathcal{S}_j$.
15:     Forward candidates $y_j^{(k)}$ with masks $M^{(k)}$. {Eq. (10)}
16:     Majority voting on $y_j^{(k)}$ and get $y_j$.
17: **end for**
18: **Return** Generated sequence $y$
---

Each $y_j^{(k)}$ serves as a *candidate prediction* for the next text token, with the final token $y_j$ selected via majority voting among the $K$ masked inputs. In case of a tie, we choose the prediction from the forward pass with the fewest dropped tokens, as it retains the most information and is deemed more reliable. By forming an ensemble of predictions derived from various subsets of the visual input, enabled through token dropout, we diversify the model's perspective on the visual content. This diversity mitigates the impact of any single misinterpretation, ultimately leading to more reliable and robust generation, which is also observed in other ensemble-based methods [24, 55, 56, 57, 58].

## 6 Experiments

We evaluate the proposed DROPOUT DECODING from two aspects: OH reduction and overall generation quality. For OH, we use the CHAIR [33] and THRONE [34] metrics to assess the performance of different decoding methods on the MSCOCO dataset. Additionally, we employ MMBench [35] to evaluate the overall generation quality and general ability of these methods.

### 6.1 Experimental Setup

**Base LVLMs.** We evaluate all methods on three representative LVLMs: LLaVA-1.5 [49], Instruct-BLIP [59] and LLaVA-NEXT [16]. LLaVA-1.5 and LLaVA-NEXT use hundreds to thousands of visual tokens for detailed representation, while InstructBLIP employs just 32 tokens but with higher information density. This showcases the flexibility of our approach, effective across models with varying token counts.

**Hallucination reduction baselines.** In addition to the original LVLM outputs, we compare our method with beam search as well as two state-of-the-art decoding methods: VCD [45], which contrasts original and distorted visuals to reduce hallucinations, and OPERA [44], which applies penalties and token adjustments for better grounding.

### 6.2 CHAIR

CHAIR [33] is a benchmark for evaluating object hallucination in image captioning. It includes two metrics: the sentence-level CHAIR$_S$, measuring the frequency of captions with hallucinated objects, and the object-level CHAIR$_I$, calculating the proportion of hallucinated objects among all objects.

**Results.** As shown in Table 1, DROPOUT DECODING consistently outperforms baseline approaches across various models, demonstrating its reliability and effectiveness in image captioning. Especially on InstructBLIP, CHAIR$_I$ and CHAIR$_S$ improve by 16% and 12% respectively over the second-best method. Furthermore, DROPOUT DECODING reduces the generation of hallucinated objects without compromising the inclusion of relevant objects. These improvements align with expectations that token dropout reduces generated objects.

| Model | Method | CHAIR | | THRONE | | | |
|---|---|---|---|---|---|---|---|
| | | $CHAIR_S \downarrow$ | $CHAIR_I \downarrow$ | $F_{all}^1 \uparrow$ | $F_{all}^{0.5} \uparrow$ | $P_{all} \uparrow$ | $R_{all} \uparrow$ |
| LLaVA-1.5 | Greedy | $42.20_{\pm 2.86}$ | $12.83_{\pm 0.36}$ | $0.795_{\pm 0.006}$ | $0.784_{\pm 0.009}$ | $0.772_{\pm 0.015}$ | $0.847_{\pm 0.010}$ |
| | Beam Search | $46.33_{\pm 1.10}$ | $13.9_{\pm 0.60}$ | $0.790_{\pm 0.007}$ | $0.772_{\pm 0.004}$ | $0.759_{\pm 0.003}$ | $\mathbf{0.862}_{\pm 0.009}$ |
| | OPERA | $41.47_{\pm 0.92}$ | $12.37_{\pm 0.72}$ | $0.802_{\pm 0.003}$ | $0.791_{\pm 0.004}$ | $0.782_{\pm 0.009}$ | $0.854_{\pm 0.011}$ |
| | VCD | $49.20_{\pm 0.88}$ | $14.87_{\pm 0.47}$ | $0.786_{\pm 0.012}$ | $0.771_{\pm 0.017}$ | $0.759_{\pm 0.020}$ | $0.854_{\pm 0.015}$ |
| | DROPOUT DECODING | $39.80_{\pm 2.3}$ | $\mathbf{11.73}_{\pm 0.25}$ | $\mathbf{0.804}_{\pm 0.002}$ | $\mathbf{0.796}_{\pm 0.006}$ | $0.790_{\pm 0.009}$ | $0.851_{\pm 0.005}$ |
| | DROPOUT DECODING (w/o prelim) | $\mathbf{39.73}_{\pm 2.15}$ | $12.20_{\pm 0.70}$ | $0.799_{\pm 0.002}$ | $0.794_{\pm 0.004}$ | $\mathbf{0.791}_{\pm 0.007}$ | $0.843_{\pm 0.005}$ |
| InstructBLIP | Greedy | $27.87_{\pm 1.32}$ | $7.90_{\pm 0.63}$ | $0.809_{\pm 0.001}$ | $0.826_{\pm 0.003}$ | $0.832_{\pm 0.006}$ | $0.803_{\pm 0.007}$ |
| | Beam Search | $25.87_{\pm 2.77}$ | $6.93_{\pm 0.569}$ | $0.809_{\pm 0.002}$ | $0.827_{\pm 0.006}$ | $0.836_{\pm 0.005}$ | $0.807_{\pm 0.015}$ |
| | OPERA | $28.07_{\pm 1.75}$ | $8.23_{\pm 0.53}$ | $0.805_{\pm 0.004}$ | $0.824_{\pm 0.003}$ | $0.830_{\pm 0.004}$ | $0.798_{\pm 0.008}$ |
| | VCD | $39.33_{\pm 2.70}$ | $19.10_{\pm 0.30}$ | $0.737_{\pm 0.008}$ | $0.746_{\pm 0.012}$ | $0.751_{\pm 0.020}$ | $0.757_{\pm 0.007}$ |
| | DROPOUT DECODING | $\mathbf{24.53}_{\pm 1.26}$ | $\mathbf{6.63}_{\pm 0.65}$ | $\mathbf{0.814}_{\pm 0.008}$ | $\mathbf{0.833}_{\pm 0.004}$ | $\mathbf{0.838}_{\pm 0.002}$ | $\mathbf{0.808}_{\pm 0.016}$ |
| | DROPOUT DECODING (w/o prelim) | $26.2_{\pm 2.40}$ | $7.10_{\pm 0.854}$ | $0.807_{\pm 0.008}$ | $0.823_{\pm 0.006}$ | $0.827_{\pm 0.010}$ | $0.804_{\pm 0.010}$ |
| LLaVA-NEXT | Greedy | $28.80_{\pm 2.12}$ | $8.10_{\pm 0.92}$ | $0.815_{\pm 0.012}$ | $0.832_{\pm 0.009}$ | $0.830_{\pm 0.007}$ | $0.799_{\pm 0.008}$ |
| | Beam Search | $28.06_{\pm 1.30}$ | $\mathbf{7.10}_{\pm 0.20}$ | $0.816_{\pm 0.007}$ | $0.834_{\pm 0.006}$ | $0.834_{\pm 0.004}$ | $0.801_{\pm 0.002}$ |
| | OPERA | $29.06_{\pm 1.89}$ | $8.06_{\pm 1.07}$ | $0.814_{\pm 0.011}$ | $0.832_{\pm 0.011}$ | $0.831_{\pm 0.006}$ | $0.799_{\pm 0.007}$ |
| | VCD | $33.19_{\pm 0.52}$ | $8.10_{\pm 0.91}$ | $0.818_{\pm 0.004}$ | $0.822_{\pm 0.003}$ | $0.808_{\pm 0.005}$ | $\mathbf{0.822}_{\pm 0.003}$ |
| | DROPOUT DECODING | $\mathbf{26.26}_{\pm 2.4}$ | $7.39_{\pm 0.69}$ | $\mathbf{0.821}_{\pm 0.010}$ | $\mathbf{0.840}_{\pm 0.009}$ | $\mathbf{0.842}_{\pm 0.002}$ | $0.805_{\pm 0.010}$ |
| | DROPOUT DECODING (w/o prelim) | $27.0_{\pm 1.80}$ | $7.53_{\pm 0.643}$ | $0.814_{\pm 0.009}$ | $0.835_{\pm 0.007}$ | $0.837_{\pm 0.003}$ | $0.793_{\pm 0.008}$ |

Table 1: Comparison of methods on $CHAIR_S$, $CHAIR_I$, $F_{all}^1$, $F_{all}^{0.5}$, $P_{all}$, and $R_{all}$ metrics for LLaVA-1.5, InstructBLIP, and LLaVA-NEXT. Details of the experimental setup and the interpretation of the standard deviation can be found in the appendix. Details of the experimental setup and the interpretation of the standard deviation can be found in the appendix.

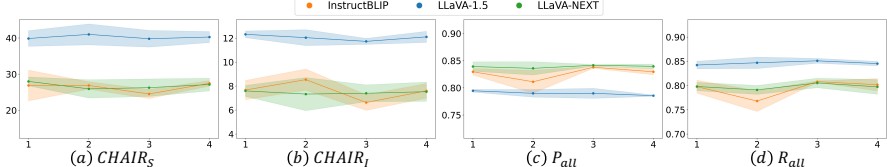

Figure 3: Comparison of $CHAIR_S$, $CHAIR_I$, $P_{all}$ and $R_{all}$ scores with standard deviations across different candidate numbers.

### 6.3 THRONE

THRONE [34] assesses hallucinations in LVLM-generated responses, covering both "Type I" (mentions of non-existent objects, like CHAIR) and "Type II" (accuracy of object existence, like POPE [43]). It uses $P_{all}$ (Precision), $R_{all}$ (Recall), $F_{all}^1$, and $F_{all}^{0.5}$. Additionally, it employs $F_\beta$, which combines $P_{all}$ and $R_{all}$, with the parameter $\beta$ controlling the weight of $R_{all}$ relative to $P_{all}$: $F_{all}^\beta = (1 + \beta^2) \cdot \frac{P_{all} \times R_{all}}{(\beta^2 \times P_{all}) + R_{all}}$.

**Results.** The test results in Table 1 illustrate that DROPOUT DECODING surpasses nearly all baseline methods across various metrics, highlighting its effectiveness in reducing both Type I and Type II hallucinations. Specifically, DROPOUT DECODING demonstrates notable strengths in InstructBLIP, excelling in the $P_{all}$ metric and achieving the highest performance in $R_{all}$. Across models, $P_{all}$ metric achieves larger improvement while the $R_{all}$ score also exceeds that of the Greedy method, confirming that retaining overlap tokens effectively preserves relevant objects. The large increase in $F_{all}^{0.5}$ further shows its comprehensiveness.

### 6.4 MMBench

MMBench [35] is a comprehensive benchmark designed to evaluate the multimodal capabilities of LVLMs across various tasks and data types. Since the prompt length limits in MMBench exceed InstructBLIP's token allowance, we report results only on LLaVA-1.5 and LLaVA-NEXT.

**Results.** As shown in Table 2, DROPOUT DECODING outperforms all the other baselines on LLaVA-1.5, which demonstrates its robustness and adaptability across a broader range of multimodal tasks.

| Method | Original | VCD | OPERA | DROPOUT DECODING |
|---|---|---|---|---|
| LLaVA-1.5 | 71.86 | 72.35 | 73.86 | **74.01** |
| LLaVA-NEXT | **74.57** | 69.65 | 74.54 | 74.31 |

Table 2: Results on MMBench. Higher is better.

## 7 Analysis and Ablation Studies

### 7.1 Efficiency Analysis

We conducted a thorough analysis of computational overhead, measuring throughput and wall-time to evaluate efficiency. Table 7.1 summarizes these results. Our method introduces additional overhead

primarily in two aspects: (1) a preliminary forward pass for identifying relevant visual tokens, (2) performing $K$ parallel forward passes using varied dropout masks.

The preliminary forward pass, though beneficial, is optional. Omitting it results in only approximately 7% throughput reduction compared to greedy decoding, while still consistently improving performance metrics across benchmarks. Furthermore, the method efficiently handles the $K$ parallel passes by batching identical inputs with distinct dropout masks into a single batched operation, significantly reducing additional computational overhead.

In terms of GPU memory, we verified efficiency under realistic conditions. Using vLLM and LLaVA-1.5 on 4×A800 80GB GPUs, GPU memory usage was 38.12 GB with efficient KV caching, unchanged between greedy decoding and our method without preliminary passes. Confirmatory experiments under Huggingface Transformers similarly demonstrated minimal GPU memory increase (from 14.02 GB to 15.31 GB), indicating negligible impact on inference constraints.

The cost of computing uncertainty metrics is explicitly included in our benchmarks and remains negligible. With LLaVA-1.5 and 576 image tokens, computing uncertainty adds only 73.30 ms per input, a minor cost amortized across the batched forward passes.

Overall, our approach effectively balances efficiency and performance.

| Metric | Greedy | Ours w/o prelim | Ours w/ prelim |
|---|---|---|---|
| Throughput (tok/s) ↑ | 37.0 | 34.1 (-7.8%) | 20.1 (-45.7%) |
| Wall-time (per 50 tok) ↓ | 1.35 | 1.47 (+8.9%) | 2.49 (+84.4%) |

Table 3: Computational overhead analysis.

## 7.2 Parallel Dropouts Hyperparameters

As in §5.2, we generate $K$ candidate predictions using token dropout. This section examines how varying the hyperparameters impacts generation quality. We fix $\delta^{(k)} = 0.1$ and adjust $\gamma^{(k)}$ based on a predefined order: $\gamma^{(1)} = 0.3$, $\gamma^{(2)} = 0.5$, and $\gamma^{(3)} = 0.7$. However, setting $\gamma^{(4)} = 0.9$ excessively drops visual tokens and degrades InstructBLIP's performance, so we set $\gamma^{(4)} = 0.1$. Moreover, our majority voting favors candidates with fewer dropped tokens in ties. To avoid identical outputs when comparing only two candidates, we remove Candidate 1 in the second round.

As shown in Fig. 3 (a) and (b), both CHAIR$_S$ and CHAIR$_I$ scores peak at $K = 3$ for LLaVA-1.5 and InstructBLIP. Increasing $K$ to 4 introduces a less-masked candidate that slightly negatively impact our method's effectiveness in reducing hallucinations. Conversely, using fewer candidates (e.g. , only Candidate 1/2) lacks the balance needed for stable voting outcomes, resulting in increased randomness. Similarly, Fig. 3 (c) and (d) shows that THRONE's $R_{all}$ and $P_{all}$ metrics also perform best at $K = 3$. Overall, we find that selecting three candidates strikes the optimal balance between increased certainty from additional votes and the controlled uncertainty introduced by candidate dropout probability, allowing DROPOUT DECODING to achieve more trustworthy and stable generation results.

## 7.3 Preliminary Forward Pass

As discussed in §5.2, DROPOUT DECODING may employ a preliminary forward pass to retain most relevant objects during generation, which helps reduce hallucinated objects while maintaining high-quality outputs. In contrast, bypassing this step risks masking relevant visual tokens during the token dropout phase, potentially degrading overall performance. However, incorporating a preliminary forward pass roughly doubles the computational cost per generation. Specifically, our goals are: 1) to confirm the effectiveness of the preliminary forward pass, and 2) to explore a more efficient alternative when computational resources are limited.

As shown in Table 1, including the preliminary forward pass consistently improves most metrics, with particular notable gains in the $F_{all}$ score on THRONE. Interestingly, for LLaVA-1.5 and LLaVA-NEXT, the variant without the preliminary pass still outperforms other baselines in most metrics. We hypothesize that this discrepancy arises from differences in the abundance of visual tokens, as LLaVA-1.5 and LLaVA-NEXT have hundreds or thousands of visual tokens. While InstructBLIP only has 32, making each token's contribution more critical. Consequently, omitting the preliminary forward pass in InstructBLIP risks losing critical information, lowering performance. These findings

suggest that while a preliminary forward pass is highly beneficial for LVLMs, models with more tokens may achieve better efficiency and performance by skipping this step.

## 7.4 Necessity of Uncertainty Guidance on Masking

As discussed in §5.1, DROPOUT DECODING incorporates epistemic uncertainty in the masking process. To validate the necessity of this approach, we compare it with a random masking strategy, which replaces uncertainty with a random method. As shown in table Table 4, although random masking performs better on the CHAIR metric, it struggles with BLEU and fails to compute the THRONE metric. We find that random masking causes repetitive token generation (e.g., "*apple apple apple...*"), artificially inflating the CHAIR score. This happens because random masking disrupts contextual information, leading to faulty generation. In contrast, our uncertainty-based approach selectively masks uncertain tokens, preserving the context and ensuring more coherent and accurate sequences.

| Model | Method | CHAIR | | BLEU | THRONE |
|---|---|---|---|---|---|
| | | $\text{CHAIR}_S \downarrow$ | $\text{CHAIR}_I \downarrow$ | BLEU $\uparrow$ | THRONE $\uparrow$ |
| LLaVA-1.5 | Greedy | $42.20_{\pm 2.86}$ | $7.90_{\pm 0.63}$ | $11.62_{\pm 0.09}$ | Table 1 |
| | Uncertainty-guided | $39.80_{\pm 2.3}$ | $11.73_{\pm 0.25}$ | $11.64_{\pm 0.12}$ | Table 1 |
| | Random | $35.93_{\pm 0.90}$ | $8.57_{\pm 0.63}$ | $11.51_{\pm 0.12}$ | Error |
| InstructBLIP | Greedy | $27.87_{\pm 1.32}$ | $7.90_{\pm 0.63}$ | $12.70_{\pm 0.54}$ | Table 1 |
| | Uncertainty-guided | $24.53_{\pm 1.26}$ | $6.63_{\pm 0.65}$ | $12.30_{\pm 0.18}$ | Table 1 |
| | Random | $23.80_{\pm 1.28}$ | $5.07_{\pm 0.71}$ | $10.88_{\pm 0.08}$ | Error |

Table 4: Comparison of masking strategies on $\text{CHAIR}_S$, $\text{CHAIR}_I$, BLEU and THRONE metrics for LLaVA-1.5, InstructBLIP.

## 7.5 High-Confidence Token Masking Analysis

To further investigate the robustness of our uncertainty-guided masking, we conducted an additional ablation experiment focusing on the opposite condition—masking high-confidence tokens instead of low-confidence ones.

Following the identical experimental setup as in Table 1 (our main CHAIR and THRONE evaluation), we replaced low-confidence masking with high-confidence masking. The results are summarized in Table 7.5.

| Model | $\text{CHAIR}_S \downarrow$ | $\text{CHAIR}_I \downarrow$ | $F_{\text{all}} \uparrow$ | $F_{0.5,\text{all}} \uparrow$ | $P_{\text{all}} \uparrow$ | $R_{\text{all}} \uparrow$ |
|---|---|---|---|---|---|---|
| LLaVA | 41.62 | 12.00 | 0.798 | 0.784 | 0.774 | 0.858 |
| LLaVA-NEXT | 27.61 | 7.82 | 0.803 | 0.823 | 0.828 | 0.789 |
| InstructBLIP | 29.50 | 9.01 | 0.808 | 0.825 | 0.825 | 0.794 |

Table 5: Effect of masking high-confidence tokens on CHAIR and THRONE metrics.

As shown, the results are generally worse than masking low-confidence tokens (i.e., masking high-uncertainty tokens as in our proposed method) and remain close to the greedy decoding baseline. This suggests that dropping high-confidence tokens has limited influence on generation quality—these tokens correspond to regions where the model is already certain, typically associated with background or redundant patches. Consequently, their removal produces only minor perturbations.

In contrast, masking low-confidence tokens directly influences the model's generative process, as these tokens are uncertain yet potentially informative (often corresponding to salient or ambiguous visual regions). Masking them introduces meaningful variability, thereby improving robustness and reducing hallucination frequency. This further validates our uncertainty-guided masking strategy as both effective and theoretically grounded.

## 8 Conclusion

We introduce DROPOUT DECODING, a novel uncertainty-guided context selective decoding approach aimed at enhancing the reliability of LVLMs. After quantifying the uncertainty in visual inputs, DROPOUT DECODING accordingly drops out visual tokens to regularize uncertainty and employs an ensemble-based decoding approach to stabilize generation. Extensive experiments on CHAIR, THRONE, and MMBench validate the effectiveness with consistent improvements over existing methods in both hallucination reduction and general multimodal capability.

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

# A  Details of Uncertainty Decomposition

A detailed derivation of Eq. (5):

$$
\begin{aligned}
U_{\text{total}} &= \mathbb{H}\left[q^{\text{proj}}\right] \\
&= -\sum_{y \in \mathcal{V}} q^{\text{proj}}(y) \log q^{\text{proj}}(y) \\
&= -\sum_{y \in \mathcal{V}} \left(\mathbb{E}_i\left[q_i^{\text{proj}}(y)\right]\right) \log q^{\text{proj}}(y) \\
&= \mathbb{E}_i\left[-\sum_{y \in \mathcal{V}} q_i^{\text{proj}}(y) \log q^{\text{proj}}(y)\right] \\
&= \mathbb{E}_i\left[\text{CE}\left(q_i^{\text{proj}}, q^{\text{proj}}\right)\right] \\
&= \mathbb{E}_i\left[\mathbb{H}\left[q_i^{\text{proj}}\right] + D_{\text{KL}}\left(q_i^{\text{proj}} \parallel q^{\text{proj}}\right)\right] \\
&= \mathbb{E}_i\left[U_{\text{ale}}(i) + U_{\text{epi}}(i)\right]
\end{aligned}
\tag{11}
$$

# B  Implementation Details

Our experiment is conducted on the MSCOCO 2014 test set, where we randomly sample 500 images across 3 random seeds. The average and standard deviation across these seeds are reported in our result table. The prompt used for the images is "Describe the image."

The experimental setup of DROPOUT DECODING is shown in Table 6. We set the maximum new tokens to 512 to ensure the complete generation of models, therefore achieving more reliable results from CHAIR and THRONE. In MMBench, as all questions are single-choice questions, we set the maximum new tokens to 1 for a more precise evaluation. We set other parameters in generation to greedy for more stable and repeatable results.

| Parameters | CHAIR | THRONE | MMBench |
|---|---|---|---|
|  | 512 | 512 | 256 |
| Top-k |  | False |  |
| Top-p |  | 1 |  |
| Temperature $\tau$ |  | 1 |  |
| Number Beams |  | 1 |  |

Table 6: Parameter settings used in our experiments.

In addition to general generation settings, DROPOUT DECODING includes hyperparameters specified in §5.2. The details of these hyperparameter settings are provided below:

**Top-$k$ in identifying relevant visual tokens.** Before the decoding process, we first obtain $q^{\text{proj}}$, which is then used in the decoding process for generating the relevant visual tokens. The higher the top-$k$ is, the more visual tokens are expected to be kept during the decoding process. In LLaVA-1.5, we set $k = 5$, and in InstructBLIP, we set $k = 10$. The difference of $k$ between LLaVA-1.5 and InstructBLIP derives from the informative level of each visual token, where in LLaVA-1.5, each visual token carries less information than in InstructBLIP, which only contains 32 visual tokens.

**Number of mask $K$.** $K$ refers to the number of predictions that will join the majority vote progress. We set $K = 3$ in our experiment settings.

**$\gamma^{(k)}$ and $\delta^{(k)}$ in uncertainty-guided masking** We set $\delta^{(k)} = 0.1, \gamma^{(k)} = 0.2 * k + 0.1; k = 1, 2, ..., K; K = 3$ in our experiment settings.

Moreover, we provide the hyperparameter settings of our baselines. OPERA's hyperparameters can be referred to Table 7; VCD's hyperparameters can be referred to Table 8.

| Parameters | Value |
|---|---|
| Self-attention Weights Scale Factor $\theta$ | 50 |
| Attending Retrospection Threshold | 15 |
| Beam Size | 3 |
| Penalty Weights | 1 |

Table 7: OPERA hyperparameter settings.

| Parameters | Value |
|---|---|
| Amplification Factor $\alpha$ | 1 |
| Adaptive Plausibility Threshold | 0.1 |
| Diffusion Noise Step | 500 |

Table 8: VCD hyperparameter settings.

## C Details of Masked Tokens

We measured the average masked visual tokens of the selected candidate in decoding, demonstrating the effectiveness of our method. For reference, the total number of visual tokens varies across models: LLaVA uses 576 visual tokens, InstructBLIP processes 32 visual tokens, while LLaVA-NEXT typically generates around 2,000 tokens depending on the resolution and patching strategy.

|  | LLaVA | LLaVA-NEXT | InstructBLIP |
|---|---|---|---|
| **Masked Tokens** | 244.14 | 879.53 | 13.67 |

Table 9: Average number of masked tokens in the selected candidate

## Impact Statement

This paper aims to advance the field of Machine Learning. While our work has potential societal implications, we do not identify any immediate concerns that require specific attention.

## Limitations

While DROPOUT DECODING improves reliability by suppressing uncertain visual tokens during inference, our method has several limitations. First, the additional forward pass in decoding, which introduces computational overhead. Second, the effectiveness of epistemic uncertainty estimation relies on the quality of text-space projections, which may be influenced by the quality of projector.

# D Generation Examples and Analysis

## D.1 LLaVA-1.5 Examples

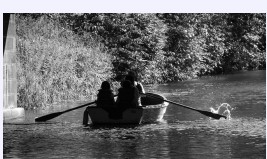

*Generated Caption:*

**Greedy Generated caption:**

The image features a black and white photograph of a small boat with three people rowing it down a river. The boat is being pulled by the people, and there are two swans swimming alongside the boat. The swans are swimming in the water near the boat, creating a peaceful and serene scene.

**DROPOUT DECODING Generated caption:**

The image features a group of people in a small boat, paddling down a river. There are at least four people visible in the boat, with some of them sitting closer to the front and others towards the back. The boat is filled with people, and they seem to be enjoying their time together on the water.

- - - - - - - - - - - - - - - - - - - - - - - - - - - - - - - - - - - - - - - - - -

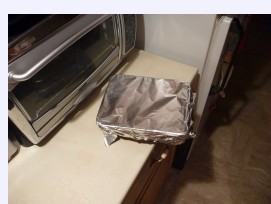

*Generated Caption:*

**Greedy Generated caption:**

The image features a kitchen countertop with a microwave oven and a large, wrapped object sitting on top of it. The object is wrapped in aluminum foil, which is covering a large rectangular box or container. The microwave oven is positioned on the left side of the countertop, and the wrapped object is placed on the right side. The countertop is also adorned with a few utensils, such as a spoon and a fork.

**DROPOUT DECODING Generated caption:**

The image features a kitchen counter with a metal pan or tray sitting on top of it. The pan is covered in aluminum foil, which is commonly used to keep food warm or to prevent it from sticking to the pan. The kitchen counter appears to be a part of a larger countertop, and the scene suggests that the pan is being prepared for cooking or baking.

- - - - - - - - - - - - - - - - - - - - - - - - - - - - - - - - - - - - - - - - - -

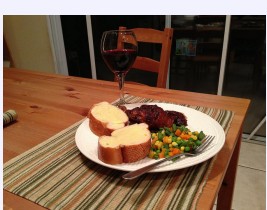

*Generated Caption:*

**Greedy Generated caption:**

The image features a dining table set with a white plate containing a variety of food items, including meat, vegetables, and bread. The plate is placed on a wooden table, and there is a glass of wine next to it. The table is set for a meal, with a fork and knife placed on the table. The food items on the plate are arranged in a visually appealing way.

**DROPOUT DECODING Generated caption:**

The image features a dining table with a white plate containing a delicious meal. The plate is topped with a variety of food items, including a piece of meat, a roll, and a side of vegetables. There are also a couple of carrots on the plate. In addition to the plate, there is a wine glass placed on the table, suggesting that the meal is accompanied by a glass of wine. A chair is positioned near the table.

## D.2 InstructBLIP Examples

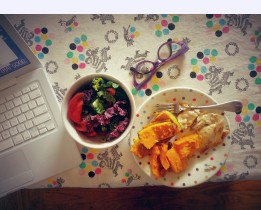

*Generated Caption:*

**Greedy Generated caption:**

The image features a table with a laptop and a plate of food, including a bowl of salad and a plate of food. The food is arranged on the table, with a fork and a knife nearby. The laptop is placed on the table, likely used for work or entertainment. The table is set up for a comfortable dining experience, with a variety of food options available.

**DROPOUT DECODING Generated caption:**

The image features a plate of food on the table is aesthetically pleasing and well-balanced. There is a plate of food, including a salad, and a bowl of chicken, and a fork. The plate is placed on the table, which is also adorned with a laptop and a computer. The laptop is placed on the table, and the computer is positioned to the left of the plate.

- - - - - - - - - - - - - - - - - - - - - - - - - - - - - - - - - - - - - -

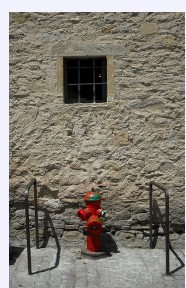

*Generated Caption:*

**Greedy Generated caption:**

The image features a red fire hydrant placed in front of a stone wall. The fire hydrant is positioned in the middle of the scene, with a window above it. There are two benches placed near the fire hydrant, one on the left and another on the right side of the image. The benches are positioned close to the fire hydrant, with one closer to the window and the other further away.

**DROPOUT DECODING Generated caption:**

The image features a red fire hydrant is placed in front of a stone building, with a window above it. The window is located on the second floor of the building, and the fire hydrant is positioned in front of the window. There are two metal bars on either side of the fire hydrant, which are likely used to secure the hydrant in place.

- - - - - - - - - - - - - - - - - - - - - - - - - - - - - - - - - - - - - -

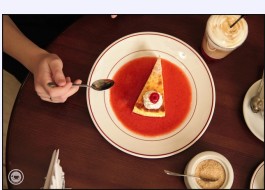

*Generated Caption:*

**Greedy Generated caption:**

The image features a dining table with a white plate topped with a slice of pie, a spoon, and a cherry on top. The pie is surrounded by various utensils, including a fork, knife, and spoon. There are also two cups on the table, one on the left and one on the right, each containing a beverage. The dining table is set with a variety of dishes, including a bowl, a cup, and a spoon. The scene is set for a delicious meal, with the pie and beverages adding to the overall ambiance.

**DROPOUT DECODING Generated caption:**

The image features a dining table is set with a variety of dishes, including a plate with a slice of pie, a cherry, and a spoon. There are also several cups of coffee and a bowl of soup. The table is set for a meal, and the dishes are arranged in a way that suggests a person is about to enjoy their food.

*Generated Caption:*

**Greedy Generated caption:**

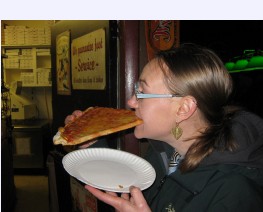

In the heart of a bustling pizza shop, a woman stands, her attention captivated by the slice of pizza she holds in her hand. The pizza, with its golden crust and vibrant toppings, is a tantalizing sight. She's dressed casually, her green jacket contrasting with the warm tones of the shop.The shop itself is a feast for the eyes. A large refrigerator stands in the background, its surface adorned with various signs and posters. The words "We guarantee the best pizza" are prominently displayed, a testament to the shop's commitment to quality. ...

**DROPOUT DECODING Generated caption:**

In the heart of a bustling pizza shop, a woman stands, her attention captivated by the slice of pizza she holds in her hand. The pizza, with its golden crust and vibrant toppings, is a tantalizing sight. She's dressed casually, her green jacket contrasting with the warm tones of the shop.The shop itself is a feast for the eyes. Boxes of pizza are neatly stacked on shelves, their white and red colors standing out against the darker tones of the shop. A sign on the wall proudly declares "We guarantee the best pizza in town", a testament to the shop's confidence in its offerings. ...

- - - - - - - - - - - - - - - - - - - - - - - - - - - - - - - - - - - - -

*Generated Caption:*

**Greedy Generated caption:**

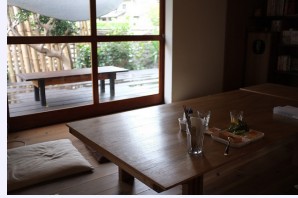

The image depicts a cozy indoor setting with a focus on a wooden table. The table is set with a few items, including a plate of food, a glass of water, and a small bowl. There are also a couple of utensils, a fork and a knife, placed on the table. The table is positioned in front of a large window that offers a view of a patio area with a bamboo fence and a small table. The room has a warm and inviting atmosphere, with a wooden floor and a beige cushion visible in the foreground. The lighting suggests it might be daytime. There are no visible texts or distinctive brands in the image. The style of the image is a candid, real-life photograph with a focus on the table setting and the view outside the window.

**DROPOUT DECODING Generated caption:**

The image depicts a cozy indoor setting with a focus on a wooden table. The table is set with a few items, including a plate with a small portion of food, a glass of water, and a pair of chopsticks. There are also a couple of small bowls and a spoon on the table. The table is positioned in front of a large window that offers a view of a wooden deck outside. The deck is furnished with a bench and a small table, and there are bamboo plants and a tree visible in the background. The room has a warm and inviting atmosphere, with a wooden floor and a cushion on the floor to the left of the table. The lighting suggests it might be daytime, and the overall style of the image is natural and serene.

### D.4 Analysis

In our examples, we use red to denote hallucinations. Specifically, DROPOUT DECODING addresses hallucinations in two key ways: eliminating non-existent objects and correcting the generation of misidentified objects. For instance, in the third example of LLaVA-1.5, the Greedy method hallucinates a non-existent object, a knife, whereas DROPOUT DECODING effectively removes this hallucination. Similarly, in the second example of InstructBLIP, the Greedy method misidentifies metal bars as benches, but DROPOUT DECODING successfully corrects this, accurately recognizing the metal bars.

