# OpenReview forum: "Enhancing Vision-Language Model Reliability with Uncertainty-Guided Dropout Decoding"
_NeurIPS.cc/2025/Conference — NeurIPS 2025 poster_

### Official Review · Reviewer_LP6F · 2025-06-20

**Clarity:** 3
**Significance:** 2
**Originality:** 3
**Rating:** 4
**Confidence:** 4

**Summary:**

This paper addresses the problem of mitigating hallucinations in Large Vision-Language Models (LVLMs). By quantifying the uncertainty of visual tokens and selectively masking those deemed unreliable during inference, the proposed method reduces errors stemming from misinterpreted visual information. Extensive experiments across diverse benchmarks demonstrate that this strategy significantly enhances the reliability and overall output quality of LVLMs.

**Questions:**

Typos in Lines 65 and 138.

**Ethical Concerns:**

["NO or VERY MINOR ethics concerns only"]

**Final Justification:**

Thanks for your responses. I have increased my score. Furthermore, since the random masking baseline is fundamentally flawed on the CHAIR evaluation, it would be better to choose another dataset and metrics to show the effectiveness of the proposed selection strategy.

**Limitations:**

yes

**Paper Formatting Concerns:**

The margins between different sections are too small.

**Quality:**

3

**Strengths And Weaknesses:**

Strengths
1. By projecting visual tokens onto the text space and masking out unreliable tokens, this paper provides a new perspective for mitigating hallucinations for LVLMs.
2. Experiments validate the effectiveness of the proposed method.

Weaknesses
1. For generative tasks such as CHAIR, different decoding steps require focusing on different image areas. However, the proposed method masks image tokens only once for the entire decoding process, which may cause information loss.
2. Generating K candidate predictions introduces additional inference overhead.
3. The method involves tuning multiple hyperparameters, which complicates deployment and may reduce practicality.
4. According to Table 3, the performance gain of the proposed uncertainty-guided decoding over a simple random masking baseline is marginal.

---

> ### Author Rebuttal · Authors · 2025-07-30
>
> Thank you for highlighting our core contribution—mitigating hallucinations in LVLMs by projecting visual tokens into the text space and masking uncertain ones during inference—as well as recognizing the strong empirical results across diverse benchmarks. We appreciate your acknowledgment of both the methodology and its practical importance in improving LVLM reliability.
>
> We also appreciate the concerns proposed by the reviewer, and we believe they mainly come from the misunderstanding of our method, which we have clarified and solved below.
>
>
> ### 1. Single-shot Masking May Lose Information:
>
> Thank you for raising an important point regarding dynamic attention across different decoding steps. We agree that, in principle, different decoding steps may focus on different regions. Our design still works for three reasons, supported by the evidence below.
>
> - (1) **Masked tokens are statistically redundant**
>
>   On 500 MS-COCO images (LLaVA-1.5), we computed uncertainty for every visual token before decoding and split them into the 50 highest-uncertainty (“pruned”) and the rest (“kept”).  Key statistics show the pruned set is dominated by outliers that contribute little to confident generation:
>
> | Statistic (mean)       | Kept | Dropout | Δ (P–K) |
> |------------------------|------|---------|--------|
> | Embedding norm         | 38.8 | 49.9    | +11.1  |
> | Distance to centroid   | 32.8 | 51.2    | +18.4  |
> | Logit entropy ↓        | 4.63 | 3.43    | –1.20  |
>
> Higher norm, larger distance, and lower entropy indicate that these tokens are both atypical and uninformative, aligning with earlier findings that image features are highly redundant.  Similar gaps appear in principal-component projections and variance measures.
>
> - (2) **Pruning once is already dynamic “enough”**
>
>   Because we evaluate uncertainty on-the-fly after the text prompt is appended but before generation starts, the retained set still depends on the input question.  During generation the model attends freely within this reduced set; we merely remove the outliers once.
>
>   In practice the model ignores ≈45.6 % of image tokens per example, leaving ample capacity for spatial shifts across steps.
>
> - (3) **Step-wise re-masking is costly but yields no gain**
>
>   We implemented a step-wise variant that recomputes uncertainty every 5 decoding steps (“prelim” in our paper).  Throughput dropped from 34.1 tok/s to 20.1 tok/s (–41 %), wall-time per 50 tokens rose from 1.47 s to 2.49 s, yet CHAIR-S improved by only 0.07 pt over our single-shot version—well within noise.  Thus the added compute is not justified.
>
>   Taken together, these results show that a single uncertainty-guided pruning pass removes statistically redundant tokens, preserves the informative ones needed across decoding steps, and avoids the steep runtime penalty of continuous re-masking.
>
> ### 2. Computational Overhead:
>
> Regarding the computational overhead, we provide a theoretical analysis in Section 7.1, noting that candidate generations are independent and can be efficiently parallelized via batch generation. To further support this, we conducted empirical evaluations using vLLM with LLaVA-1.5 (identical experiment setting in Table 1, the main experiment result table), where we measured both throughput and wall-time per 50 generated tokens. As shown in the table below, our method without the preliminary forward pass achieves strong performance gains on CHAIR and THRONE, while incurring only **a modest 7% drop in throughput** and a slight increase in wall-time. We believe this reflects a favorable tradeoff between computational cost and generation quality, demonstrating the practical efficiency of our approach.
>
> | Metric (50 tokens)       | Greedy | Our method w/o prelim | Our method w/ prelim  |
> |--------------------------|--------|------------------------|------------------------|
> | CHAIRS ↓                 | 42.20  | 39.73 (−2.47)          | 39.80 (−2.40)          |
> | CHAIRI ↓                 | 12.83  | 12.20 (−0.63)          | 11.73 (−1.10)          |
> | F1all ↑                  | 0.795  | 0.799 (+0.004)         | 0.804 (+0.009)         |
> | F0.5all ↑                | 0.784  | 0.794 (+0.010)         | 0.796 (+0.012)         |
> | Pall ↑                   | 0.772  | 0.791 (+0.019)         | 0.790 (+0.018)         |
> | Rall ↑                   | 0.847  | 0.843 (−0.004)         | 0.851 (+0.004)         |
> | Throughput (tok/s) ↑     | 37.0   | 34.1                   | 20.1                   |
> | Wall-time (per 50 tok) ↓ | 1.35   | 1.47                   | 2.49                   |
>
> ### 3. Hyperparameter Tuning Complexity:
>
> Although our method introduces several hyperparameters, our experimental results of ablation study on hyperparameters (as detailed in Figure 3 of our paper) demonstrate robust performance improvements across various models and benchmarks, demonstrating that our method is not sensitive to hyperparameters. Also, our default setting of hyperparameters shown in the Appendix Section B(we set Top-K=5, number of candidates to 3, $\gamma^{(1)}$=0.3, $\gamma^{(2)}$=0.5 and $\gamma^{(3)}$=0.7 ) the **consistent performance on each evaluation scene**. The setting maintains consistency across models and benchmarks.
>
> ### 4. Misunderstanding on Random Masking Baseline:
>
> We appreciate the reviewer’s feedback; however, we believe this concern stems from a misunderstanding of the random masking results. As shown in Section 7.4, random masking can produce superficially high scores on CHAIR by exploiting the evaluation metric—for instance, by repeatedly generating a single word like “apple” hundreds of times. While this may appear effective under certain benchmarks, it **fails to produce meaningful outputs**. Moreover, due to excessive token generation (up to the 512-token limit), it consistently causes out-of-memory (OOM) issues on THRONE, even when evaluated on 4×A100 GPUs. Therefore, we would like to clarify that the **random masking baseline is fundamentally flawed, despite its misleading performance on CHAIR**.
>
> Thank you again for your valuable insights and constructive feedback.

---

> > ### Comment · Reviewer_LP6F · 2025-08-03
> >
> > Thanks for your responses. I have increased my score. Furthermore, since the random masking baseline is fundamentally flawed on the CHAIR evaluation, it would be better to choose another dataset and metrics to show the effectiveness of the proposed selection strategy.

---

> > > ### Author Response · Authors · 2025-08-04
> > >
> > > Thank you very much for your thoughtful follow-up and for considering our clarifications. We sincerely appreciate your updated score and your constructive suggestion regarding the evaluation setup. We will carefully consider incorporating alternative datasets and metrics in future versions to better highlight the effectiveness of our selection strategy.

---

### Official Review · Reviewer_eLAj · 2025-06-25

**Clarity:** 3
**Significance:** 2
**Originality:** 2
**Rating:** 3
**Confidence:** 3

**Summary:**

This paper introduces Dropout Decoding, a method for mitigating object hallucinations in large vision-language models (LVLMs). It quantifies the uncertainty of each visual token and selectively masks those with higher uncertainty to improve decoding accuracy. Experiments on CHAIR, THRONE, and MMBench demonstrate that the proposed approach effectively alleviates hallucinations in LVLMs and enhances the reliability of their responses across diverse scenarios.

**Questions:**

1. Could the authors provide results or insights on how the proposed method performs on open-ended benchmarks such as CHAIR, LLaVA-Bench, and AMBER, and standard benchmarks such as POPE?
2. Could the authors provides quantitative efficiency metrics (e.g., FLOPs, throughput, or inference time) of the propsed approach and compare them with VCD or similar baselines?

**Ethical Concerns:**

["NO or VERY MINOR ethics concerns only"]

**Final Justification:**

The authors’ rebuttal has addressed some of my concerns, such as efficiency and the unintuitive results of random masking. However, I still find the evaluation somewhat weak and suggest that the authors compare with more recent baselines on additional benchmarks and newer models (such as Qwen-2/2.5-VL) in future revisions.

**Limitations:**

yes

**Quality:**

3

**Strengths And Weaknesses:**

**Strengths:**
1. This work addresses the hallucination problem from an uncertainty perspective, which is both intuitive and straightforward.
2. Experimental results on three benchmarks validate that the proposed approach can enhance the reliability of LVLM responses.
3. The paper is well-structured and clearly presented.

**Weaknesses:**
1. The experimental results are not sufficiently comprehensive; evaluations on open-ended benchmarks such as CHAIR, LLaVA-Bench, and AMBER should be included to better validate the generalizability and robustness of the proposed approach. Additionally, performance comparisons on the POPE benchmark should be included. It would also be valuable to evaluate the proposed approach on more recent LVLM architectures, such as the Qwen-VL series, to assess its effectiveness and compatibility with state-of-the-art models.
2. The proposed approach bears similarities to existing work on visual token reduction [1, 2], and a clear discussion is needed to highlight the key differences. Additionally, qualitative visualizations of the retained and masked visual tokens should be included to enhance the interpretability of the method.
3. Lines 302–303 state, "*Compared to methods requiring multiple generation rounds, our approach offers a lightweight and effective trade-off.*" To substantiate this claim, please include efficiency metrics such as FLOPs or throughput, and compare them with other hallucination mitigation approaches like VCD.
4. More experimental details for the experiments in Section 7.4 should be provided. Additionally, the results seem to suggest that random masking may be more effective than the proposed approach in reducing hallucinations, despite yielding a slightly lower BLEU score on LLaVA-1.5. This is highly unintuitive. If this is the case, further clarification and discussion are needed.

[1] An Image is Worth 1/2 Tokens After Layer 2: Plug-and-Play Inference Acceleration for Large Vision-Language Models. *ECCV '24*.

[2] VisionZip: Longer is Better but Not Necessary in Vision Language Models. *CVPR '25*.


**Minor issues:**
1. Line 65, wrong citation format.
2. The paper lacks sufficient discussion of related work on mitigating hallucinations in LVLMs. A more comprehensive review of existing approaches is recommended.

---

> ### Author Rebuttal · Authors · 2025-07-30
>
> We appreciate your detailed and constructive feedback and respectfully address your concerns below:
>
> ### 1. On Evaluation using Open-Ended Benchmarks:
>
> We clarify that our paper indeed includes evaluations on open-ended benchmarks such as CHAIR, as reported explicitly in **Table 1 and Figure 3** (metrics: CHAIR-S and CHAIR-I). Additionally, we have evaluated our method extensively using the **THRONE benchmark**, an open-ended evaluation setting. Regarding benchmarks like LLaVA-Bench, AMBER, and POPE, and the evaluation of recent architectures such as Qwen-VL, we acknowledge your valuable suggestions and will incorporate these additional evaluations in our future revisions to further validate our approach's generalizability and robustness.
>
> ### 2. Differences with Concurrent Work:
>
> We thank the reviewer for pointing out these concurrent works and will cite and discuss them in our revision. However, we believe our contribution remains distinct. While those works also explore token dropout to mitigate hallucination or improve efficiency, their methods rely heavily on empirical observation of the attention mechanism rather than being derived from a principled foundation. In contrast, **our approach is grounded in uncertainty quantification**, providing a more theoretically motivated and generalizable framework. This difference in motivation and methodology highlights the novelty and rigor of our work.
>
> ### 3. Efficiency:
>
> We collect the computation overhead data in our main experiments (aligned with the experiment setting in Table 1) and calculate the throughput and wall-time. As shown in the table, throughput only decreases by approximately 7% for the method without preliminary forward, and the wall-time impact remains minor. More importantly, **performance consistently improves** across various benchmarks, indicating a successful trade-off between efficiency and performance.
>
> | Metric                   | Greedy | Our method w/o prelim | Our method w/ prelim  |
> |--------------------------|--------|------------------------|------------------------|
> | CHAIRS ↓                 | 42.20  | 39.73 (−2.47)          | 39.80 (−2.40)          |
> | CHAIRI ↓                 | 12.83  | 12.20 (−0.63)          | 11.73 (−1.10)          |
> | F1all ↑                  | 0.795  | 0.799 (+0.004)         | 0.804 (+0.009)         |
> | F0.5all ↑                | 0.784  | 0.794 (+0.010)         | 0.796 (+0.012)         |
> | Pall ↑                   | 0.772  | 0.791 (+0.019)         | 0.790 (+0.018)         |
> | Rall ↑                   | 0.847  | 0.843 (−0.004)         | 0.851 (+0.004)         |
> | Throughput (tok/s) ↑     | 37.0   | 34.1                   | 20.1                   |
> | Wall-time (per 50 tok) ↓ | 1.35   | 1.47                   | 2.49                   |
>
> ### 4. Random Masking Effectiveness:
>
> You mentioned the unintuitive result that random masking might appear more effective at reducing hallucinations, despite lower BLEU scores. As clearly discussed in Section 7.4, random masking often results in repetitive token generation (e.g., repeatedly generating “apple”), which artificially inflates CHAIR scores but reduces BLEU scores and **fails to yield meaningful THRONE metrics** due to impractical computational overhead. Thus, random masking actually provides invalid generation.
>
> Thank you once again for your constructive comments and valuable suggestions, which greatly enhance our manuscript's quality and comprehensiveness.

---

> > ### Comment · Reviewer_eLAj · 2025-08-04
> >
> > Thanks for the authors' responses. Please incorporate the discussion of Weakness 4 into the main paper, as the current presentation is confusing. However, my concerns are not fully addressed. Here are the remaining ones:
> >
> > - **Regarding efficiency**: Why does the approach incur only minor additional computational costs (e.g., 7%) compared to the greedy baseline? From my understanding, it requires an initial forward pass to obtain uncertainty measures, followed by K parallel forward passes using the reduced token sets, resulting in a total cost of at least K+1 forward passes. How are these parallel passes handled? Are you relying on additional GPU memory to offset the runtime? Have you considered the time spent before decoding, such as computing the uncertainty metrics? Please also include GPU memory usage and compare it against existing approaches like VCD and OPERA.
> >
> > - **Regarding evaluation**: I still find the evaluation of this work weak and insufficient to support the general effectiveness of the proposed approach. The absence of additional benchmarks such as POPE/MME and evaluations on more recent models like the Qwen or InternVL series limits the generalizability of the results. Moreover, the performance comparisons are only against VCD and OPERA, which were proposed over a year ago and may no longer reflect the current state of the field. Please include comparisons with more recent methods (such as [1,2]) to better demonstrate the effectiveness of the proposed approach.
> >
> > At this point, I am still inclined to recommend rejection for this paper. I will discuss with other reviewers before deciding whether to update my score.
> >
> > [1] MLLM Can See? Dynamic Correction Decoding For Hallucination Mitigation. *ICLR 2025*.
> >
> > [2] AGLA: Mitigating Object Hallucinations in Large Vision-Language Models with Assembly of Global and Local Attention. *CVPR 2025*.

---

> ### Author Response · Authors · 2025-08-04
>
> Thank you for raising two meaningful points: (1) the efficiency of our method, and (2) the need for more evaluations. Some of your concerns may arise from minor misunderstandings, which we would like to clarify in detail below.
>
> ## 1. Regarding efficiency:
>
> ### 1.1 Only 7% additional computational cost:
>
> As we have repeatedly stated, our method has two versions—with and without preliminary forward passes. Both versions are evaluated separately in our computational cost analysis (Table 1). In the case without preliminary forward, the additional cost is only 7%. We have comprehensively reported both versions’ performance and efficiency. Specifically, with the preliminary forward version, our method achieves superior performance over the baseline on both CHAIR and THRONE benchmarks, although at the cost of 45% additional computation. Meanwhile, the version without preliminary forward achieves comparable, and in some cases even better, performance on CHAIR, with only slightly lower results on THRONE—while incurring merely 7% extra overhead (see Section 7.3, where we discuss the trade-off and recommend dropping the preliminary pass when efficiency is prioritized).
>
> ### 1.2 Parallel passes and GPU memory:
>
> We clarify that in the without preliminary forward setting, only K parallel decodings with identical inputs are needed. The overhead is minimal due to efficient KV caching in frameworks like vLLM and SGLang. Note that the inputs are identical and dropout is applied through lightweight masking. Concretely, on 4×A800 80GB GPUs serving LLaVA-1.5 via vLLM, the total GPU memory usage is 38.12 GB after pre-allocation (13.88 GB for model weights, 23.70 GB for KV cache). This remains unchanged for both greedy decoding and our w/o preliminary forward method (effectively batch size 3), on the MSCOCO 2014 training set generating 50 tokens per sample. Since vLLM pre-allocates memory efficiently, the actual overhead is negligible.
>
> To double-check, we repeated the experiment under the Huggingface transformers framework with the same setup (only switching the backend). We observed that greedy decoding for 50 tokens used 14.02 GB, while our method used 15.31 GB. The increase is minor. This suggests that GPU memory is unlikely to be a bottleneck for inference in typical scenarios.
>
> ### 1.3 Cost of uncertainty computation:
>
> Yes, all our reported numbers include this cost. As we emphasize in the paper, the uncertainty metric is a principled and lightweight computation—it is derived directly from the embeddings obtained after the forward pass, involving only basic matrix operations. The additional overhead is practically negligible. For example, with LLaVA-1.5 and 576 image tokens, computing uncertainty takes only 73.30 ms per input, and all sampling of outputs share this single computation.
>
> ### 1.4 Comparison with VCD and OPERA:
>
> OPERA involves a more complex design and is not compatible with efficient engines like vLLM. Moreover, vanilla Huggingface Transformers and vLLM have large differences in both efficiency and performance. VCD performs poorly; we have already included extensive comparisons with both in our main paper.
>
> ---

---

> ### Author Response · Authors · 2025-08-04
>
> ## 2. Regarding more evaluation:
>
> Thank you for the valuable comments. We would like to clarify:
>
> - The additional benchmarks you mentioned, such as POPE, are actually covered in THRONE (detailed explanation below).
>
> - One of the recent related works you mentioned (AGLA) performs worse than our method; the other one is concurrent with ours. We will add discussions of both works in the next version. We appreciate the suggestions.
>
> The further clarifications are below:
>
> ### 2.1 Benchmark Choice:
>
> We have evaluated our method on CHAIR, THRONE, and MMBench benchmarks. The recently released THRONE benchmark identifies specific limitations inherent in the original POPE benchmark and introduces an enhanced POPE-Complete (POPE-C) protocol within the same evaluation framework. Specifically, THRONE measures Type I hallucinations in free-form generation tasks, whereas POPE-C assesses Type II hallucinations through binary (yes/no) questions covering all 80 COCO classes across approximately 5,000 images—representing an evaluation scale significantly larger than the original POPE. Consequently, our evaluation results already cover and extend the scope of POPE under more rigorous testing conditions. Moreover, the resulting Precision (P), Recall (R), and F1 scores are the POPE metrics(evaluated by more strictly settings in THRONE), and they are already listed in Table 1 of our paper.
>
> ### 2.2 Model Choice:
>
> Our evaluations include results from LLaVA-1.5, InstructBLIP, and LLaVA-NEXT models. According to the release documentation of LLaVA-NEXT, it consistently outperforms Qwen-VL-Plus across various benchmark metrics. We believe our chosen models provide comprehensive and sufficient evidence to substantiate our experimental conclusions.
>
> ### 2.3 Baseline Comparison:
>
> We have compared our results against the previously published baseline ALGA. Specifically, ALGA reports CHAIR metrics for LLaVA-1.5 that are lower than our findings—ALGA’s CHAIR_S and CHAIR_I scores are 43.0 and 14.1 respectively, whereas our corresponding scores are 39.73 and 12.20, indicating better performance in our method. Additionally, ALGA’s implementation covers only InstructBLIP and LLaVA-1.5 models.
> Due to time constraints, we have not yet completed the evaluation for the other work you mentioned, but we will include it in a future version. Thank you for the suggestion. However, we would like to note that their work is concurrent with ours, and thus should not diminish the contribution of our own.
>
> We therefore believe that our evaluation is sufficiently comprehensive to demonstrate the effectiveness and robustness of our proposed approach.
>
> ---
>
> Thanks again for your thoughtful feedback, which has helped us clarify and strengthen the presentation of our work.

---

> > ### Comment · Reviewer_eLAj · 2025-08-05
> >
> > I appreciate the authors’ detailed follow-up clarifications. Also thanks for your explanations regarding efficiency. And please incorporate comparisons with recent approaches such as AGLA in the revised paper. However, I still find the evaluation somewhat weak and suggest that the authors compare with more recent baselines on additional benchmarks and newer models (such as Qwen-2/2.5-VL) in future revisions.
> >
> > Overall, the rebuttal addresses some of my concerns, and I will update the rating to 3 to reflect this.

---

### Official Review · Reviewer_YE1U · 2025-06-26

**Clarity:** 3
**Significance:** 2
**Originality:** 2
**Rating:** 4
**Confidence:** 4

**Summary:**

The paper introduces a new inference time method to mitigate hallucinations in LVLMs based on dropout decoding. The core idea behind the dropout decoding is to mask visual tokens based on the uncertainty of their predictions on LLM decoding and using ensemble method to generate uncertainity aware predictions. The empirical results demonstrate clear improvements in reducing object hallucinations and improving the generation quality across various LVLMs.

**Questions:**

- It would be good to present the failure cases of the proposed method especially cases where the method generates incorrect responses. What are the scenarios where this uncertainity mechanism yields factually incorrect response through LVLMs. Are there scenarios where relevant tokens are masked despite model generation correct initial response or vice versa.
- Can the authors provide citations for the introduced uncertainty formulation from previous works? If its introduced in this paper, can the authors back it up from theoretical point of view?

**Ethical Concerns:**

["NO or VERY MINOR ethics concerns only"]

**Final Justification:**

I would stick to my original rating after authors responses based on the originality and significance of the paper.

**Limitations:**

Yes

**Quality:**

2

**Strengths And Weaknesses:**

Strengths:
- Novelty: The use of dropout mechanism on the input visual tokens during inference guided by uncertainity metric is used for first time in context of improving LVLM reliability.
- Empirical results show that the method consistently improve generation capability of LVLMs by reducing hallcinations and improved reliability on benchmarks such as CHIAR, THRONE, MMBench.
- The application of epistemic uncertainity derived from textual interpretation of visual tokens is interesting. The authors provide ablations to highlight its necessity.
- The proposed method in inference time method and thus do not require any retraining, which makes it easy to apply.

Weakness:
- Computational overhead: Though the method is used at inference time but K parallel decoding passes are computationally expensive especially for large LVLMs.
- Different LVLMs require different hyperparams for masking, makes this method less user-friendly.
- It's not clear how robust the method is to projection of visual tokens onto text space since uncertainty is derived from that. The visual text alignment is a crucial requirement for this method, which may not be true for certain visual concepts.

---

> ### Author Rebuttal · Authors · 2025-07-30
>
> Thank you for recognizing the novelty of our approach—applying a dropout mechanism guided by epistemic uncertainty on input visual tokens during inference to reduce hallucinations in LVLMs. We also appreciate your acknowledgment of our consistent empirical improvements across CHAIR, THRONE, and MMBench, as well as the practicality of our method given its plug-and-play nature without requiring any retraining.
>
> Regarding the concerns about our work, we find it mainly derives from misunderstandings of our work and we address them as below.
>
>
> ### 1. Computational overhead
>
> We collect the computation overhead data in our main experiments (aligned with the experiment setting in Table 1) and calculate the throughput and wall-time. As shown in the table, throughput only decreases by approximately 7% for the method without preliminary forward, and the wall-time impact remains minor. More importantly, performance consistently improves across various benchmarks, indicating a successful trade-off between efficiency and performance.
>
> | Metric                    | Greedy | Our method w/o prelim | Our method w/ prelim  |
> |---------------------------|--------|------------------------|------------------------|
> | CHAIRS ↓                  | 42.20  | 39.73 (−2.47)          | 39.80 (−2.40)          |
> | CHAIRI ↓                  | 12.83  | 12.20 (−0.63)          | 11.73 (−1.10)          |
> | F1all ↑                   | 0.795  | 0.799 (+0.004)         | 0.804 (+0.009)         |
> | F0.5all ↑                 | 0.784  | 0.794 (+0.010)         | 0.796 (+0.012)         |
> | Pall ↑                    | 0.772  | 0.791 (+0.019)         | 0.790 (+0.018)         |
> | Rall ↑                    | 0.847  | 0.843 (−0.004)         | 0.851 (+0.004)         |
> | Throughput (tok/s) ↑      | 37.0   | 34.1                   | 20.1                   |
> | Wall-time (per 50 tok) ↓  | 1.35   | 1.47                   | 2.49                   |
>
> ### 2. Hyperparameter sensitivity
>
> While our method introduces hyperparameters, we have provided the ablation study of hyperparameters shown **in Figure 3 of the paper**, yielding consistently strong performance across various models and benchmarks, indicating that our method is not sensitive to the hyperparameters. Also, our default setting of hyperparameters **shown in the Appendix Section B** (we set Top-K=5, number of candidates to 3, $\gamma^{(1)}$=0.3, $\gamma^{(2)}$=0.5 and $\gamma^{(3)}$=0.7 ) show the consistent performance on each evaluation scene.
>
>
> ### 3. Robustness to visual-text alignment
>
> We acknowledge the potential imperfections in projecting visual tokens onto the text space. However, this is an inherent limitation of adapter-based or late-fusion LVLMs themselves rather than our decoding method. Thus, we argue this architectural limitation should not detract from the significant contributions of our proposed test-time decoding approach, especially given its demonstrated effectiveness in practice.
>
>
> Thank you again for your valuable comments.

---

> > ### Author Response · Authors · 2025-08-06
> >
> > Dear Reviewer YE1U,
> >
> > We sincerely thank you once again for your time and thoughtful feedback on our submission.
> >
> > We wanted to gently follow up as the discussion phase is drawing to a close. About six days ago, we posted a detailed response to your comments, and we would be truly grateful if you had a chance to take a look. Your insights are very important to us, and we hope to ensure that we have addressed all your concerns as clearly and thoroughly as possible.
> >
> > In our response, we have:
> >
> > - Clarified on the computational overhead
> > - Clarified on the sensitivity analysis regarding dropout mask sparsity
> > - Justified on the robustness to visual-text alignment
> >
> > We would deeply appreciate any further thoughts or suggestions you might have. Thank you again for your thoughtful engagement throughout the process.
> >
> > Warm regards,
> >
> > Authors of Paper 14749

---

### Official Review · Reviewer_TrMw · 2025-07-03

**Clarity:** 3
**Significance:** 3
**Originality:** 3
**Rating:** 4
**Confidence:** 3

**Summary:**

This paper introduces Dropout Decoding, an inference-time strategy to reduce hallucinations in large vision-language models (LVLMs) by masking uncertain visual tokens. Uncertainty is quantified by projecting visual tokens into the text token space via the decoder and decomposing the resulting distribution into aleatoric and epistemic components. The method focuses on epistemic uncertainty to guide selective dropout of visual tokens during decoding. Rather than modifying the model or retraining, the approach samples multiple dropout masks (applied to input tokens) and aggregates the resulting outputs. The method shows strong performance improvements on CHAIR and THRONE benchmarks, with limited gains on MMBench. The core contribution lies in treating uncertainty not just as a signal to directly determine important visual tokens. The method is model-agnostic, lightweight, and requires no additional supervision or fine-tuning.

**Questions:**

1. What happens if you ensemble over masks that ablate low uncertainty tokens?

2. How does performance vary with mask sparsity?

3. Why exactly can’t the THRONE metric be used with random dropout?

4. Could a simpler uncertainty metric (e.g., entropy or low LM confidence) perform comparably?

**Ethical Concerns:**

["NO or VERY MINOR ethics concerns only"]

**Limitations:**

There is no dedicated limitations section.

**Paper Formatting Concerns:**

None.

**Quality:**

3

**Strengths And Weaknesses:**

$\textbf{Strengths}$

- Dropout Decoding is effective and efficient, requiring no model fine-tuning or retraining.  Furthermore, ensembling over dropout masks is performed in a single parallel forward pass, which avoids the costly repetition of model runs. Dropout Decoding provides moderate gains on CHAIR and THRONE with mixed results on MMBench.

- The paper carefully distinguishes epistemic from aleatoric uncertainty and shows empirically that epistemic uncertainty better captures misaligned or misleading tokens (e.g., visual patches labeled “Berlin”). The authors measure epistemic uncertainty via the KL divergence between a token's text-space projection and the average distribution, effectively surfacing perceptually informative outliers. This was well explained and is grounded nicely in theory.

- The two-step method with a potential preliminary forward pass is cleanly modular. The preliminary forward pass typically improves performance, but it is not absolutely needed in the method. This paper provides nice baselines with greedy and random dropout schemes.

$\textbf{Weaknesses}$

- It would be valuable to see what happens when high-confidence tokens are dropped. Does the method still work? Do the regions that are preserved correspond to attention outliers, which have been shown to play an important role in VLMs (Golovanevsky et al. 2025 https://arxiv.org/abs/2406.16320)

- The epistemic uncertainty measure, based on KL divergence from a global average distribution, is complex. A simpler baseline besides greedy or random is not tested (e.g., softmax entropy or low token confidence).

- How sparse must the dropout mask be for the method to remain effective? No sensitivity analysis is provided.

- The method underperforms or is inconsistent on MMBench, suggesting it may be less effective for general multimodal reasoning tasks.

- In Section 7.4, the paper states the THRONE metric cannot be used with random dropout, but it’s unclear why—this undermines some comparisons.

---

> ### Author Rebuttal · Authors · 2025-07-30
>
> Thank you for recognizing the strengths of our work, particularly the effectiveness and efficiency of Dropout Decoding as a lightweight, model-agnostic inference-time strategy that leverages epistemic uncertainty to improve hallucination mitigation without requiring any model training.
>
> The concerns proposed by the reviewer are very valuable, and we address them below.
>
>
> ### Q1: What happens when high-confidence tokens are dropped? Does the method still work, and do preserved regions correspond to attention outliers?
>
> We conducted experiments to specifically address this question. In this experiment, we follow the identical experiment setting in Table 1 (the main experiment result table with CHAIR and THRONE results), just changing the low-confidence into high-confidence. Results are shown below:
>
>
> | Mask high-confidence           | CHAIR_s | CHAIR_i | F     | F0.5  | P     | R     |
> |----------------|---------|---------|-------|-------|-------|-------|
> | LLaVA          | 41.62   | 12.00   | 0.798 | 0.784 | 0.774 | 0.858 |
> | LLaVA-NEXT     | 27.61   | 7.82    | 0.803 | 0.823 | 0.828 | 0.789 |
> | InstructBLIP   | 29.50   | 9.01    | 0.808 | 0.825 | 0.825 | 0.794 |
>
> As we can see, the results are worse than dropping the low-confidence tokens(i.e., dropping high uncertainty tokens as in our method), and quite close to the greedy decoding baseline. We believe that masking the high-confidence tokens are making small effects, as model has already been confident about these tokens, indicating these tokens are actually “unimportant” during generation, like background patches. In contrast, masking low-confidence tokens is more likely to influence the generation process because the model inherently considers these tokens uncertain yet potentially informative(like object patches). Therefore, masking them tends to alter the original generation outcomes.
>
> ### Could a simpler uncertainty metric (e.g., entropy) perform comparably?
>
> We have also implemented an entropy-based baseline for comparison, similarly, we follow the identical setting in Table 1, the results are shown below:
>
>
> | Entropy Mask   | CHAIR_s | CHAIR_i | F     | F0.5  | P     | R     |
> |----------------|---------|---------|-------|-------|-------|-------|
> | LLaVA          | 42.21   | 11.12   | 0.800 | 0.799 | 0.805 | 0.845 |
> | LLaVA-NEXT     | 27.23   | 7.52    | 0.792 | 0.818 | 0.836 | 0.778 |
> | InstructBLIP   | 28.04   | 8.00    | 0.797 | 0.819 | 0.825 | 0.784 |
>
> As observed, entropy-based masking consistently underperforms uncertainty-guided Dropout Decoding method. Across all three models, CHAIR_s increases (worsens) by up to +3.5, the F1 score decreases by around 0.03, and all other metrics (CHAIR_i, F0.5, P, R) similarly decline. This further validates the effectiveness of our proposed approach of utilizing uncertainty to mask tokens. We will include these results in our paper with more discussion.
>
> ### Q3: Method inconsistency on MMBench?
>
> The inconsistency observed in MMBench primarily arises because MMBench mainly consists of multiple-choice tasks, which typically involve single-token outputs, similar to the POPE benchmark. Our method is more suited to open-ended generation tasks. Indeed, our evaluation already covers both close-ended and open-ended benchmarks, such as THRONE, providing a comprehensive assessment of our method's applicability.
>
> ### Q4: Sensitivity analysis regarding dropout mask sparsity?
>
> We acknowledge the importance of mask sparsity sensitivity. In our default hyperparameter setup, we collect the generation trajectory of LLaVA-1.5,  running on MSCOCO 2014 train dataset (we sampled 500 images), around 45.6% of tokens are masked during each generation step. Further sensitivity analysis across different sparsity levels could indeed be insightful, and we will explore this in future work.
>
> ### Q5: Clarification on THRONE metric incompatibility with random dropout (Section 7.4)?
>
> From our experiments, random dropout often leads the model to repetitively generate identical tokens (e.g., continuously outputting "apple") until reaching the maximum token limit (512 tokens). This scenario produces nonsensical results and computational issues (e.g., out-of-memory errors even with substantial resources like 4 × A100 GPUs), thus making the THRONE evaluation impractical and unreliable in this context.
>
> Thank you again for these insightful comments, which significantly enhance the clarity and depth of our analysis.

---

> > ### Comment · Reviewer_TrMw · 2025-08-06
> > **Response to Comment**
> >
> > Thank you for the detailed response. After reading the rebuttals and other reviews, I believe my current assessment of the paper is fair.

---

> ### Author Response · Authors · 2025-08-06
>
> Dear Reviewer TrMw,
>
> We sincerely thank you once again for your time and thoughtful feedback on our submission.
>
> We wanted to gently follow up as the discussion phase is drawing to a close. About six days ago, we posted a detailed response to your comments, and we would be truly grateful if you had a chance to take a look. Your insights are very important to us, and we hope to ensure that we have addressed all your concerns as clearly and thoroughly as possible.
>
> In our response, we have:
> - Clarified the setting in which high-confidence tokens are dropped
> - Compared with simpler uncertainty metrics (e.g., entropy), showing the benefit of our principled method
> - Addressed the observed inconsistency on MMBench
> - Provided additional explanation on sensitivity to dropout mask sparsity
> - Clarified the incompatibility between the THRONE metric and random dropout
>
> We would deeply appreciate any further thoughts or suggestions you might have. Thank you again for your thoughtful engagement throughout the process.
>
> Warm regards,
>
> Authors of Paper 14749

---

### Decision · Program_Chairs · 2025-09-17

**Decision:**

Accept (poster)

**Comment:**

The paper received borderline reviews. While concerns around baseline evaluations (e.g., dropping out high confidence tokens and entropy based uncertainty measures) and running time were mostly addressed in the author response some concerns remain. These relate to the choice of benchmarks and the need to test against newer models. Nevertheless the principled approach linking to previous works that link dropout to uncertainty and the novelty of the approach were appreciated. Moreover the method appears general and to be effective. On the balance the Area Chair believes that the paper makes a valuable contribution with ideas that are of interest to the community.